# Israel and the Apostolic Mission: A Post-Supersessionist Reading of Ephesians and Colossians

Lionel J. Windsor

Moore Theological College, Sydney 2042, Australia; lionel.windsor@moore.edu.au

**Abstract:** Interpretation of Ephesians and Colossians has often proceeded on the basis that the stance of the original authors and recipients towards Israel is supersessionist, i.e., that the church has entirely replaced or superseded Israel as the locus of divine scriptural promises. By contrast, this article presents a post-supersessionist reading of Ephesians and Colossians. The reading strategy seeks to read the letters as situated within the dynamics of the apostolic mission to proclaim the gospel of Jesus as the Jewish *christos*/messiah to the nations. This mission is envisaged in Acts as a priestly dynamic in which the blessings of salvation in the *christos*/messiah began within a distinctly Israelite original community and proceeded to the nations without necessarily negating Jewish distinctiveness. The reading highlights key instances of this Israel-centered missionary dynamic in Ephesians and Colossians. It also seeks to demonstrate how this dynamic helps to provide satisfactory answers to key exegetical questions in the letters. Furthermore, it offers alternative non-supersessionist readings of critical passages concerning circumcision, law, and Jewish identity in the two letters. The article is a distillation and summary of research in the author's previously published book *Reading Ephesians and Colossians After Supersessionism: Christ's Mission through Israel to the Nations*.

**Keywords:** Ephesians; Colossians; post-supersessionism; replacement theology; mission; Israel; Jew/Judean; Judaism/s; Christian/ity; New Testament

Interpreters of New Testament documents, including Ephesians and Colossians, have often proceeded on the assumption that the situation of the original authors and readers vis-à-vis Israel is identical to that of the later Christian church. In this view, the church, comprising mainly non-Jewish Christ-believers, is regarded as an entity separate from ethnocultural Israel that has entirely replaced or superseded Israel as the locus of divine scriptural promises. At times, such supersessionist views have contributed towards disastrously hostile Christian attitudes towards Jewish people (Barth 1969, pp. 45–54; Rader 1978, pp. 95–96, 203–4). Furthermore, such views risk downplaying the redemptive-historical shape of Christian origins, impoverishing contemporary Christian self-reflection.

In this article, I present a *post-supersessionist* reading of Ephesians and Colossians.[1] I am distilling and summarizing research presented in more detail in my book *Reading Ephesians and Colossians After Supersessionism: Christ's Mission through Israel to the Nations* (Windsor 2017).[2] While my reading is essentially exegetical, I seek to avoid the common tendency to rely on supersessionist assumptions when determining the most probable answers to key exegetical questions. Instead, I argue exegetically for alternative readings that are not informed or controlled by such supersessionist assumptions.

Existing non-supersessionist readings of the New Testament, Ephesians and Colossians in particular,[3] have arisen in various disciplines, contexts, and conversations. These include progressive dispensationalist interpretation (e.g., Hoch 1992), constructive biblical theology (e.g., Robinson 1996, pp. 113–15; 2008, pp. 105–8; cf. Goldsworthy 2012, pp. 164–65, 201–6), theologically oriented commentary (e.g., Barth 1974; Fowl 2012),[4] scholarship informed by the New Perspective on Paul (e.g., Yee 2004), scholarship informed by the Paul within Judaism perspective (e.g., Allen 2018; Campbell 2008), Messianic Jewish interpretation (e.g., Kinzer 2005, pp. 165–71; 2015, pp. 65–82), and explicit post-supersessionist theological reflection

(e.g., Soulen 1996). Several insights from these existing interpretations inform my reading. I recognize that Ephesians and Colossians exhibit a non-totalizing vision of unity in the *christos*/messiah that allows for ongoing diversity, assume that Jewish distinctiveness is assigned a positive value despite the existence of Jew-gentile hostility, take the letters as allowing for an ongoing priority for Israel within God's purposes through Jesus as the *christos*/messiah, and affirm a gentile *christos*-believing identity that is both distinct from and connected to Israel. Nevertheless, I do not temper the strongly Christocentric focus apparent in Ephesians and Colossians nor downplay its implications for the present transformation of both Jewish and gentile identity in Jesus.

In particular, my post-supersessionist reading is informed by the recognition that in Ephesians (explicitly) and Colossians (implicitly), Israel's distinct identity is conceived in terms of a priestly vocation towards the world, which is inextricably linked to the apostolic mission to preach Jesus as the Jewish *christos*/messiah to the nations (see esp. Kinzer 2015, pp. 65–82; Robinson 2008, pp. 84–85).

## 1. An Evangelical Post-Supersessionist Reading Strategy

My method is an "evangelical post-supersessionist reading" (Windsor 2017, p. 3). This label is intended to indicate both a positive and negative aspect of the reading strategy.[5] Positively, I am seeking to read the letters as situated within the dynamics of the apostolic mission to "proclaim the evangel" or gospel (εὐαγγελίζω/εὐαγγέλιον) of Jesus as the Jewish *christos*/messiah to the nations.[6] In both Acts and Romans, this apostolic mission is explicitly framed in terms of blessings proceeding from or through Israel to others (e.g., Acts 1:8; Rom 1:16). My reading strategy entails being alert to the existence of this dynamic in Ephesians and Colossians. As a result—negatively—I deliberately call into question readings of Ephesians and Colossians that assume that Jewish distinctiveness is always a problem that needs to be overcome. Distinctiveness need not always imply hostility and exclusion; it may be understood in terms of a positive vocation towards others. Thus, Jewish distinctiveness may be connected closely with the apostolic mission in Ephesians and Colossians.

The approach may be illustrated by contrast with Lincoln's reading of Ephesians (Lincoln 1987, 1990). Lincoln strongly rejects the view that Ephesians should be located amid the Pauline mission. Instead, Lincoln argues, the primary concerns of Ephesians are situated after the dynamics of the Pauline mission have settled. Thus, the author of Ephesians envisages the issues surrounding this mission, including its concrete struggles over Jewish and gentile identity, as past issues which have largely been resolved in favor of an overarching unity in Christ (Lincoln 1987, p. 619). This means, for Lincoln, that Ephesians contains a supersessionist viewpoint in which "Israel's role is replaced by that of the church" (Lincoln 1987, p. 621). By contrast, my approach situates the concerns of Ephesians (and Colossians) *within* the Pauline mission, which allows for issues concerning Jewish distinctiveness to be more dynamic and complex.[7]

### 1.1. Positively: Situating Ephesians and Colossians within the Apostolic Mission

There is significant warrant for situating the concerns of Ephesians and Colossians within, rather than outside, the apostolic mission. Various features of the letters provide this warrant. These include explicit references to a global gospel mission of which the readers are both beneficiaries (Eph 3:1–13; Col 1:6, 23) and participants (Eph 6:18–19; Col 4:2–6); authorial self-descriptions of Paul as a missionary presently struggling and suffering to bring divine revelation to others (Eph 3:1–13; 4:1; 6:19–20; Col 1:24–29); references to the concrete communication of a message, described using the terms "gospel" (Eph 1:13; 3:6; 6:15, 19; Col 1:5, 23), "evangelize" (Eph 2:17; 3:8; cf. 4:11), and "teach" (Eph 4:21; Col 1:28; 2:7); references to "apostles" along with "prophets" as foundational missionaries (Eph 2:20; 3:5), linked with other evangelists and teachers (Eph 4:11–12; cf. Col 1:7; 4:10–11); narrative-like descriptions locating the readers within an unfolding account of the progress of the gospel mission over time using aorist indicative verbs (Eph 2:13, 17; 3:2–8; 4:11; Col 1:5–7,

23, 25; 2:7);[8] and marked descriptions of divine blessings proceeding from writer and/or associates ("we" or "I") to recipients ("you [also]") (Eph 1:12–13, 2:22; 3:2; Col 1:25–27).

The connection of these letters with the Israel-centric dynamic in the book of Acts is also evident.[9] Several lexical and structural features of Ephesians and Colossians are aligned with significant features of the description of the apostolic mission in Acts. These include narrative-like depictions of the progress of the apostolic mission in which blessings move temporally from an original group to others (Eph 1:13–14; Col 1:4–6, 23), utilizing a cluster of terminology also found in Acts to describe the progress of the gospel from Israel to the nations (Acts 10:1—11:18);[10] explicit descriptions of Israel's blessings being shared with non-Israelites (Eph 2:11–22; 3:5–6), explicitly labeled in Israel-centric terms as "the nations/gentiles" (τὰ ἔθνη) (Eph 2:11; 3:1, 6, 8; Col 1:27; cf. Acts 10:45; 11:1–3, 18);[11] marked attention drawn to the fact that gentiles can enjoy the benefits of holiness and the Spirit by faith in Jesus as the *christos alongside* Jews (Eph 2:18–22; Col 1:26–27; cf. Acts 10:45, 47; 11:17–18; 15:7–9; 26:17–18); descriptions of Jewish hostility towards gentiles linked to issues surrounding access to temple worship (Eph 2:11–22; cf. Acts 21:27–29); and the mention of "decrees" (δόγματα) in relation to the law of Moses and gentiles (Eph 2:15; cf. Acts 16:4).[12]

In addition, the mention of "apostles and prophets" in Ephesians suggests a close connection between the apostolic mission and Israelite identity. In Ephesians, "the apostles and prophets" (τῶν ἀποστόλων καὶ προφητῶν) (Eph 2:20; cf. 3:5) are foundational figures in relation to the gospel-preaching mission (cf. 2:17; 3:8).[13] In Acts, the apostles and the prophets—including Paul—are consistently connected with the original apostolic community at Jerusalem and portrayed as distinctly Israelite figures, even as they take part in the gentile mission (Acts 8:14; 11:1, 27; 12:25—13:1; 15:2, 4, 27, 32; 21:10, 17–26).

These features and parallels provide sufficient warrant to approach Ephesians and Colossians on the basis that Jewish distinctiveness may have a positive value in relation to the apostolic mission, rather than assuming that it necessarily represents a problem that must be overcome.

### 1.2. Negatively: Questioning Supersessionist Over-Readings

The negative aspect of this reading strategy involves questioning common supersessionist interpretations of specific passages in Ephesians and Colossians. I have designated these interpretations "supersessionist over-readings" because they "extrapolate from the explicit statements found in the texts to make further conclusions about race, ethnicity, or Jewish practice—conclusions that are not necessary implications of the texts themselves" (Windsor 2017, p. 29). These over-readings explain the meaning of these passages in Ephesians and Colossians in a way that resembles the clear supersessionist views found in later decades (e.g., Barnabas).[14] There are three main categories.

The first category of over-reading concerns physical circumcision. Ephesians 2:11 mentions physical circumcision while describing hostility and alienation between Jews and gentiles. Colossians 2:11–13 affirms a spiritual circumcision given to gentile believers in the *christos*. Supersessionist over-readings interpret these passages to mean that physical circumcision is entirely devoid of value for Jewish believers. Physical circumcision, in these readings, is not merely invalid for gentiles (cf. Gal 2:3); it is a marker of opposition to Christ and so invalid for Jews also (Boyarin 1994, p. 27; Bruce 1984, pp. 103–4; Calvin 1965, [ca. 1548], pp. 331–32; Lincoln 1987, pp. 609–10; 1990, p. 136).

The second category of over-reading concerns Jewish law-observance more broadly. Ephesians 2:14–15a describes how the blood of the *christos* ends Jew-gentile hostility "by abolishing the law of the commandments in ordinances." Colossians 2:13–23 describes the death of the *christos* in opposition to various entities that some interpreters link with the Mosaic law. Supersessionist over-readings interpret these passages to mean that Christ has abolished the concrete Jewish observance of the Mosaic law. This is described in various ways. Some regard the "ceremonial" aspect of the law as abolished for all, including for Jews (Aquinas 1966, pp. 105–7; Calvin 1965, [ca. 1548], pp. 151, 334–37). Others regard the boundary-marking function of the law—i.e., Jewish social practices such as circumcision

and Sabbaths that highlight Jewish distinctiveness—as abolished (Dunn 1996, pp. 171–75; Bevere 2009; Wright 1986, p. 119). Others see the entire Mosaic law as abolished, meaning that the Mosaic covenant is canceled and has no binding force on either gentiles or Jews in any sense (Arnold 2010, p. 163; Lincoln 1987, pp. 611–12; 1990, p. 142; Perkins 2000, pp. 399–400).

The third category of over-reading concerns Jewish identity itself. Ephesians 2:14–16 describes the peace-making activity of the *christos* in terms of "creat[ing] the two …into one new humanity" (2:15). Colossians 3:9–11 advocates a "new" kind of "humanity" in which "there is not Greek and Jew, circumcision and uncircumcision" (3:11). Supersessionist over-readings interpret these passages to mean that Christ has negated all forms of distinct Jewish identity. Various rationales are given. Chrysostom (1840, [ca. 390], pp. 148–52) regards Jewish distinctiveness as an obstacle to peace. Calvin (1965, [ca. 1548], pp. 151–52, 350), because of his opposition to "ceremonies," regards Paul as urging Jews to give up their distinct identity because it is an "external" condition alien to Christ. Many twentieth-century interpreters use overtly racial terms to argue that Christianity represents a "third race," replacing old ethnic identities (Best 1955, pp. 152–54; 1998, p. 269; Bruce 1984, pp. 295–96; Dunning 2006, p. 14; Gnilka 1977, p. 139; Von Harnack 1908, pp. 240–65; Hoehner 2002, pp. 379–80; Lincoln 1987, pp. 612, 616; 1990, pp. 144, 163; Martin 1991, p. 31; Talbert 2007, p. 82).[15] This view appears to be reflected in the addition of the supersessionist phrase "in place of" in the 1946 RSV translation of Eph 2:15 (retained by NRSV, ESV).[16] Other scholars state that any form of social "distinction" is regarded as a negative factor opposed to peace and freedom (Dunn 1996, p. 223; Foster 2016, pp. 48, 110; Lohse 1971, pp. 143–44).

These interpretations may appear self-evident from the perspective of later supersessionist forms of Christian theology. However, they are not obvious when viewed from an earlier perspective within the progress of the apostolic mission. As we have seen, this mission was understood as proceeding from a distinctively Israelite core to the surrounding nations without necessarily negating all forms of Jewish distinctiveness. In the rest of the article, I will present the key features of my reading of Ephesians and Colossians, seeking to demonstrate that locating the concerns of the letters within the apostolic mission progressing from Israel to the nations provides satisfactory non-supersessionist answers to key exegetical questions.

## 2. The Framing of Ephesians: A Priestly Dynamic (Ephesians 1:1, 3)

### 2.1. The Original Designation for the Addressees (1:1)

The first issue that arises in Ephesians is a textual issue concerning the original designation for the addressees (Eph 1:1). Several key witnesses read "to the holy ones who are indeed/also believers/faithful in the *christos* Jesus" (τοῖς ἁγίοις τοῖς οὖσιν καὶ πιστοῖς ἐν Χριστῷ Ἰησοῦ).[17] Other witnesses include "in Ephesus" (ἐν Ἐφέσῳ) after "who are" (οὖσιν),[18] which brings the form closer to other canonical Pauline epistles with addressees in named cities.[19] This suggests that the first reading is original since it is easy to see why a later scribe would insert "in Ephesus" but quite challenging to see why a scribe would omit the phrase.[20] However, interpreters often regard the first reading as impossible, because it appears incomprehensible. The issue is not simply the unusual syntax but the meaning itself, since "almost by definition the saints are those who are faithful in Christ Jesus" (Best 1997a, p. 5). Since it appears impossible from a modern viewpoint to explain why the author would write in such a strange and tautological way, interpreters normally either reject this reading in favor of the inclusion of "in Ephesus" (e.g., Hoehner 2002, pp. 144–48) or suggest complex prior stages in the textual history (e.g., Best 1997b; Lincoln 1990, pp. 1–4).

However, by locating the concerns of Ephesians within the apostolic mission from Israel to the nations, an evangelical post-supersessionist perspective can make sense of the first reading and explain why it may well be original (cf. Caird 1976, pp. 30–31). As Trebilco (2012, pp. 122–63) demonstrates, the phrase "the holy ones" (οἱ ἅγιοι) was most likely an eschatological self-designation for the original Israelite community in Jerusalem. As the gentile mission proceeded, the designation was also used to refer to gentile believers

in Christ. However, such usage would have been conspicuous. As we have seen, both in Acts and elsewhere in Ephesians, the idea that gentiles, not just Jews, can enjoy the benefits of holiness through faith in Jesus as the Jewish *christos* is not merely an unremarkable background assumption; instead, it is the focus of marked attention (Eph 2:18–22; Acts 10:45, 47; 11:17–18; 15:7–9; 26:17–18). So the expression "to the holy ones who are indeed/also believers in in the *christos* Jesus" need not be dismissed as tautological; instead, it may be regarded as drawing the readers' attention from the outset to the *remarkable* fact that gentiles who believe in the *christos* are included in Israel's eschatological holiness. Thus, there is no reason to reject it as the most likely original reading. The exclusion of "in Ephesus" would also imply that the letter was originally intended for a wider gentile believing audience beyond Ephesus, which is consistent with the lack of concrete details in the author's description of the recipients' situation (3:2; 4:21; 6:22).[21]

*2.2. A Priestly Dynamic of Blessing (1:3)*

The expression "Blessed be the God …" (εὐλογητὸς ὁ θεός) (Eph 1:3) introduces a form that is frequently identified as a "blessing" or *berakhah* (בְּרָכָה). This Jewish form often appears in the Scriptures and other Second Temple literature in expressions of praise to God for salvation and protection (Lincoln 1990, pp. 10–11). In Jewish *berakhoth*, God is commonly named "the God of Israel" or Israel's ancestors.[22] However, in Eph 1:3, God is named "the God and father of our lord Jesus *christos*" (ὁ θεὸς καὶ πατὴρ τοῦ κυρίου ἡμῶν Ἰησοῦ χριστοῦ). While the markedly christological focus of the phrase is often discussed (e.g., Lincoln 1990, p. 11), a less-discussed feature of the passage is also worthy of attention (Fowl 2012, pp. 39–41). God is described as "the one who has blessed us …in [the] *christos*" (ὁ εὐλογήσας ἡμᾶς ...ἐν χριστῷ). In the Jewish Scriptures, the expression "they will be blessed in …" (וְהִתְבָּרְכוּ ב/וְנִבְרְכוּ ב/ἐνευλογηθήσονται ἐν) is concentrated almost exclusively in the Genesis narratives. It is used to foreshadow a priestly dynamic in which all the nations of the earth "will be blessed in" Abraham/Isaac/Jacob and their "offspring" (Gen 12:3; 18:18; 22:18; 26:4; 28:14; cf. Sir 44:21). This phrase from Genesis is applied to the apostolic mission in both the undisputed Pauline epistles (Gal 3:8, 14, 28–29) and Acts (Acts 3:25). Therefore, the use of such "blessed in" language in Eph 1:3 also suggests an allusion to this scriptural priestly dynamic grounded in the Abraham narratives. It depicts the *christos* as Abraham's "seed." In this understanding, Jesus as the *christos* does not replace Israel; instead, he fulfills scriptural promises that Israel would be the channel of God's blessing to the nations.

## 3. The Apostolic Mission in Focus (Ephesians 1:11–14)

This identification of a priestly dynamic from the outset of Ephesians leads to a further question concerning the variation between the first-person plural "we"/"us" and the second-person plural "you" in Eph 1, especially in the transition from "we" (ἡμᾶς) (1:12) to "also you" (καὶ ὑμεῖς) (1:13).[23]

Some interpreters argue that the transition is merely stylistic (e.g., Best 1998, p. 148). However, this does not account for the markedly temporal nature of the transition: it is not simply from "we" to "you" but from "we who first hoped in the *christos*" (ἡμᾶς ...τοὺς προηλπικότας ἐν τῷ χριστῷ) (1:12) to "also you, having heard" (καὶ ὑμεῖς ἀκούσαντες) (1:13). This implies that the author at this point is deliberately drawing attention to a temporal progression from one group to another.

Other interpreters see 1:13 as marking a transition from Israel or Jewish believers (1:11–12) to gentile believers (1:13–14) (e.g., Barth 1974, vol. 1, pp. 130–33).[24] In favor of this view is the fact that "you" (ὑμεῖς) are later explicitly designated as "the gentiles/nations" (τὰ ἔθνη) as distinct from Israel (2:11). Furthermore, as we have seen, it is likely that there is already a Jew-gentile priestly dynamic implicit in the *berakhah* beginning in 1:3. However, the Jew-gentile distinction is not explicitly in the foreground at this point in the discourse. This suggests that more explanation is necessary.

An earlier view advocated by Aquinas (1966, p. 64) is worth considering. Aquinas sees 1:13 as marking a transition from "we," the apostles, to "you," the recipients. At least



two features of the passage support this view. Firstly, 1:9 refers to the revelation of a "mystery" (μυστήριον) "to us" (ἡμῖν), while later in the letter, the "mystery" (μυστήριον) (3:4) is explicitly described as having been revealed "to his holy apostles and prophets" (3:5).[25] Secondly, as we have already seen, 1:13–14 contains an explicit description of the reception of the apostolic mission, employing a cluster of terminology also found in Acts 10:1—11:18. This strongly suggests that "we" denotes the original apostolic community from whom the gospel came, and "you" denotes the recipients of the apostolic gospel.

This does not mean we are forced to choose between a Jew-gentile dynamic and an apostolic missionary dynamic since the two are related (cf. Kinzer 2015, pp. 70–72). According to Acts, the original apostolic community—including Paul—represented a renewed Israel through whom blessing came to the nations. This implies that, while the progress of the apostolic mission to preach the gospel of Jesus as the *christos* to others is in the *foreground* of Eph 1:11–14, this mission is undergirded by a priestly dynamic in which blessings progress from Israel to the nations. While this Jew-gentile progression is in the *background* in Eph 1, it becomes more prominent in the subsequent discourse.

### 4. Israel and the Nations in Focus (Ephesians 2:11–22)

*4.1. Reconciliation between Israel and the Nations as a Distinct Topic*

Ephesians 2 comprises two halves (2:1–10; 2:11–22). Both halves depict a former plight for the readers that has now been resolved. In each case, the *christos* Jesus is the focus of the resolution. In 2:1–10, the plight primarily concerns spiritual death, wrath, and alienation from God; it is resolved through being raised together to life and saved by grace. In 2:11–22, the plight primarily concerns alienation and hostility between two groups, explicitly named "the nations" and "Israel"; it is resolved through proximity, peace, and reconciliation with both God and one another. Lincoln (1990, pp. 608–10) regards the first half as the author's primary focus and the second half—which explicitly mentions Israel and the nations—as illustrative of the first and therefore subsidiary to the author's purposes. This is a crucial plank of Lincoln's argument that Israel's role in Ephesians is temporary and superseded by Christ.

However, this view does not consider critical differences between the two halves of the chapter. In the first half, the plight is described as *common* to both "you" (2:1–2) and "also we" (2:3). In the second half, the plight is described as *different* for the two groups: it is only "you, the nations" (2:11) who were "far off," whereas others were "near" (2:13, 17). Hence, the vision of the *christos*'s reconciling work in the two halves of Eph 2 cannot be reduced to a single dimension. While believers from Israel and the nations share a common history when it comes to the nature of their reconciliation with God (cf. Ezek 36—37) (Starling 2011, pp. 191–92), they have different histories when it comes to their respective paths of reconciliation with one another. While both "you" and "we" need to be made alive with the *christos*, it is only "you, the nations" who need to be "brought near" (2:11, 13). This multidimensional vision of unity-in-distinction means we should be cautious about assuming too quickly that the author regards the work of the *christos* as abolishing all distinctions entirely.

*4.2. Circumcision (2:11)*

Ephesians 2:11 includes physical circumcision in a description of Jew-gentile alienation. As we have seen, this has led some interpreters to infer that the author regards physical Jewish circumcision and the distinctiveness it implies in an entirely negative light. However, this is an unnecessary over-reading.

The problem being addressed here is not distinctiveness per se, but hostility (cf. 2:14). Many of the terms in 2:12—"*christos*" (χριστοῦ), "alienated" (ἀπηλλοτριωμένοι), "commonwealth" (πολιτείας), "covenants" (διαθηκῶν), "promise" (ἐπαγγελίας), and "hope" (ἐλπίδα)— or their cognates have parallels in Second Temple Jewish descriptions of overt hostility between Israel and the nations (e.g., Pss. Sol. 17.3, 5, 13, 15, 32; 2 Macc 13:14). This hostility was at times expressed on both sides using derogatory comments about circumcision.

Gentiles used "circumcision" as a term of derision for Jews (e.g., Philo, *Spec. Laws* 1.2; Josephus, *Ag. Ap.* 2.137) (Barclay 1996, pp. 438–39). Correspondingly, Jews used "foreskin" (ἀκροβυστία) as a term of distancing and derision against gentiles (e.g., Gen 34:14; Acts 11:3). This use of derisive terms is consistent with the phrasing used in Ephesians. The problem being highlighted here is not circumcision itself but the name-calling often associated with it: gentiles were "those **called** 'foreskin' by those **called** 'circumcision'" (οἱ **λεγόμενοι** ἀκροβυστία ὑπὸ τῆς **λεγομένης** περιτομῆς) (2:11). While this depicts circumcision as a feature used to support hostile attitudes, it does not necessarily imply that circumcision itself is being viewed negatively.

Furthermore, the description of circumcision as "in the flesh, hand-produced" (ἐν σαρκὶ χειροποιήτου, 2:11) does not empty circumcision of any value whatsoever. Rather, it forms a contrast with believers' salvation as the "product" (ποίημα) of God in the previous clause (2:10). The phrase simply highlights the fact that circumcision by itself is unable to bring about the kind of change needed to overcome the hostility since what is required is a work of God to destroy the enmity.

### 4.3. The Mosaic Law (2:14–15)

Ephesians 2:15 describes the *christos*, by his blood/cross, as "having broken down the dividing wall of the fence, the hostility, by abolishing the law of the commandments in decrees" (τὸ μεσότοιχον τοῦ φραγμοῦ λύσας, τὸν νόμον τῶν ἐντολῶν ἐν δόγμασιν καταργήσας). As we have seen, this has led some interpreters to infer that concrete Jewish observance of the Mosaic law has been abolished. Again, however, this is an unnecessary over-reading.

The term "the law" (τὸν νόμον) should not be made central to interpretation (contra Shkul 2009, pp. 79–141). This is the only place where the term appears in Ephesians, and it is qualified with genitive and prepositional modifiers (Campbell 2008, p. 16). Furthermore, the participial phrase in which it appears is syntactically subordinate to the previous phrase concerning breaking down of the dividing wall (2:14) (Yee 2004, pp. 147–48). Hence, the reference to "the law" can only be interpreted in light of the broader discourse concerning the breaking down of the dividing wall.

The architectural term used for the dividing wall (μεσότοιχον) (2:14) refers to a wall internal to a building. This implies that the author is alluding to something concrete and specific. Given the temple imagery elsewhere in the letter (e.g., 2:21–22), several interpreters see it as a reference to the balustrade in the Jerusalem temple that fenced off gentiles from the more sacred precincts reserved for Jews (Campbell 2008, p. 16; Cohick 2010, pp. 76–77; MacDonald 2004, p. 434; cf. Yee 2004, pp. 148–49). A written decree accompanied the balustrade that any gentile who passed through would be liable to a death sentence (Josephus, *Ant.* 15.417; *J.W.* 5.193–94, 6.124–26). This decree was based on commandments from the Mosaic law (see, e.g., Num 1:51).

This helps to explain the formulation: "the law of the commandments in decrees" (Kinzer 2015, p. 77). A "decree" (δόγμα) is "a formal statement concerning rules or regulations that are to be observed" in a specific situation (Bauer et al. 2000). The use of the term in Eph 2:15 is unlikely to be redundant; accordingly, the phrase is not to be seen as a reference to the Jewish observance of the Mosaic law per se. Instead, it is a more precise reference to specific applications of the Mosaic law by contemporary Jewish authorities through "decrees" that reinforced Jew-gentile hostility, especially in relation to the temple (cf. Acts 21:27–29). Thus, Eph 2:14–15a can be understood as follows: the death of the Jewish *christos*, by rendering gentiles holy (cf. 2:19, 21), has rendered invalid the law *as interpreted by decrees promoting Jew-gentile hostility*. Hence, it has metaphorically broken down the dividing wall in the temple. The literal meaning of this metaphor is that the death of the Jewish *christos* has removed Jew-gentile hostility.[26]

A parallel may be seen in Acts 15. Acts 15:1–5 describes attempts by "some" (τινες) (15:1, 5) Jews to impose strict directives concerning gentile circumcision. Presumably, they were attempting or wishing to have decrees to this effect issued. In response, the apostles issued their own "decrees" (δόγματα) (Acts 16:4) to the gentile *ekklēsiai*. These decrees were

grounded in the understanding that God had cleansed the gentiles by faith (15:9). Accordingly, the apostolic decrees promoted Jew-gentile fellowship (15:19–29). At no point in the narrative is it suggested that Jews must abandon their own law-observance, since this is not the issue.[27]

### 4.4. Jewish Identity (2:14–16)

Ephesians 2:15 describes the peace-making activity of the *christos* as "creat[ing] the two, in himself, into one new humanity" (τοὺς δύο κτίσῃ ἐν αὐτῷ εἰς ἕνα καινὸν ἄνθρωπον). As we have seen, this has led some interpreters to infer that Christ has negated all forms of distinct Jewish identity. Yet again, however, this is an over-reading. The expression does not necessarily imply the eradication of distinctions. The term "one" (εἷς) is often used to describe a unity in which distinctions continue to co-exist (see, e.g., Eph 5:31) (Woods 2014, pp. 105–13). Furthermore, while the concept of "new creation" implies renewal and transformation, it does not necessarily mean the destruction of all features of the old (e.g., Eph 4:23–24) (Woods 2014, pp. 113–22). Indeed, critical features in Eph 2:11–22 suggest that distinctions continue to characterize the new humanity, albeit in a transformed manner.

Firstly, at the start of 2:11–22, the readers are urged to "remember" their past gentile identity from an Israelite perspective, thus conceiving of themselves not simply as believers but as gentiles who have been brought near to Israel (2:11) (Fowl 2012, pp. 85–90). This is reinforced by the marked reference to the gentile believers as "also you" (καὶ ὑμεῖς) who are being built together into a dwelling-place for God (2:22).

Secondly, the repeated use of the term "both" (ἀμφότεροι) implies that duality is significant within the new humanity. This duality is not simply a feature of the readers' hostile past but of their peaceful and united present since "both" is used as the subject of a present-tense verb: "for through him *we both have* (ἔχομεν …οἱ ἀμφότεροι) access in one spirit to the Father" (2:18).

Thirdly, three *syn*-compounds—i.e., terms beginning with the prefix "with" (σύν)—are used in the passage to describe gentile believers (Campbell 2008, pp. 21–22; Hoch 1982, p. 180; Kinzer 2015, pp. 78–79). The prefix implies that gentile believers share benefits *alongside* Israel rather than being merged with or replacing Israel (cf. 2:5–6).[28] They are "fellow-citizens (συμπολῖται) of the holy ones" (2:19),[29] every construction is "being joined together" (συναρμολογουμένη) (2:21), and so they are "being built together" (συνοικοδομεῖσθε) into God's dwelling (2:22).

Hence, the focus of Eph 2 is not on the destruction of all distinctions between Israel and the nations. Instead, it is on the messianic renewal and transformation of the relationship between Israel and the nations in such a way that peace replaces hostility so that the two groups can worship God together. This conclusion is reinforced when we observe the many parallels between Ezek 37 and Eph 2 (Suh 2007).[30] Ezekiel anticipates that the two divided tribally defined kingdoms—Ephraim and Judah—will be united in worship under the peace-making activity of the messianic king, without ever suggesting that they lose their distinct identities (Ezek 37:15–28) (Suh 2007, p. 731). The unity of Jew and gentile in Eph 2:11–22 parallels this vision. It suggests that gentile believers are presented here not as replacing Israel but as fulfilling prophetic promises concerning Israel's eschatological salvation.[31]

## 5. The Apostolic Mission as a Priestly Dynamic (Ephesians 2:17–3:21)

### 5.1. Jew-Gentile Contours to the Apostolic Mission (2:17–22)

We have seen that in Eph 1:11–14, the topic of the apostolic mission to preach the gospel of Jesus as the Jewish *christos* to others was in the foreground, whereas the idea of a priestly dynamic in which blessings progress from Israel to the nations, while present, was in the background. In Eph 2:11–22, the reverse is true: the priestly dynamic in which the nations join Israel in worship becomes a topic in the foreground. Nevertheless, the theme of the apostolic mission to preach the gospel of Jesus as the Jewish *christos* to the nations

continues to be present in the background. Indeed, this theme becomes more prominent as the discourse progresses.

The apostolic mission becomes explicit in 2:17 through the description of the peacemaking activity of the *christos* in terms of a missionary dynamic (Sandnes 1991, pp. 227–29). The *christos*, through the apostolic mission, "came and preached the gospel" (ἐλθὼν εὐηγγελίσατο) (2:17). This missionary work undertaken by the "apostles and prophets"—key figures in the original Israelite apostolic community—is foundational for the inclusion of gentile believers in Israel's privileges and the establishment of temple worship (2:20).[32] The current progress of the apostolic mission is also implied by the present tense verb describing the gentile believers as "being built together" (συνοικοδομεῖσθε) into a dwelling-place for God (2:22).

*5.2. The Pauline Mission as a Priestly Dynamic (3:1–21)*

In Eph 3, the discourse shifts to address the topic of *Paul's own* activity in relation to the apostolic mission. Rather than seeing this as an "excursus" (so Barth 1974, vol. 1, p. 327) or an exercise in social identity formation for a later generation (so Esler 2007; Shkul 2009, pp. 142–72), the reading presented here enables us to regard the concerns of 3:1–13 as naturally integrated into the prior discourse. The author here takes two significant themes from the letter so far—the apostolic mission to preach Jesus as the Jewish *christos* to others and the priestly dynamic of blessing progressing from Israel to the nations—and brings them together in relation to the Pauline mission (Kinzer 2015, pp. 79–81). This can be seen in several ways.

Firstly, the beneficiaries of Paul's apostolic ministry are explicitly designated in Israel-centric terms as "the nations" (τὰ ἔθνη) (3:1, 8).[33]

Secondly, the content of "mystery" that was previously described as revealed and communicated through the apostolic mission (1:9) (see above) is now specified with three *syn*-compounds that emphasize gentile participation in Israel's benefits (Hoch 1982, pp. 180–81; Kinzer 2015, p. 80). The mystery is "that the nations are fellow-heirs, fellow-members of the body, and fellow-participants of the promise in the *christos* Jesus" (εἶναι τὰ ἔθνη συγκληρονόμα καὶ σύσσωμα καὶ συμμέτοχα τῆς ἐπαγγελίας ἐν χριστῷ Ἰησοῦ) (3:6). This sharing of blessing takes place "through the gospel" (διὰ τοῦ εὐαγγελίου) (3:6), i.e., the message proclaimed in the apostolic mission (1:13; 2:17). Furthermore, there is a prayer that the recipients may have a share in knowledge "with all the holy ones" (σὺν πᾶσιν τοῖς ἁγίοις) (3:14).

Thirdly, Paul is identified with the apostolic community using the term "holy" (ἅγιον), which, as we have seen above, carries connotations of an eschatological priestly dynamic of blessing proceeding from Israel to the nations. The mystery has been revealed "to his [God's] *holy* apostles and prophets" (τοῖς ἁγίοις ἀποστόλοις αὐτοῦ καὶ προφήταις) (3:5). Correspondingly, Paul, who elsewhere describes himself as "the least of the apostles" (ὁ ἐλάχιστος τῶν ἀποστόλων) (1 Cor 15:9), is here described more pointedly as "the very least of all [the] *holy ones*" (τῷ ἐλαχιστοτέρῳ πάντων ἁγίων) (Eph 3:8).

Finally, Paul's apostolic ministry to the nations is depicted in cosmic priestly terms, with frequent allusions to temple imagery. These include the use of words and phrases such as "access" (προσαγωγή) (3:12; cf. 2:18); the *christos*'s "dwelling" (κατοικῆσαι) (3:17; cf. 2:22); and "the breadth and length and height and depth" (τὸ πλάτος καὶ μῆκος καὶ ὕψος καὶ βάθος) (3:18; cf. Ezek 43:13–17 LXX) (Foster 2007).

**6. Implications for Gentile Readers (Ephesians 4—6)**

The evangelical post-supersessionist reading presented here can help to elucidate significant features of the second half of Ephesians (Eph 4—6). In this section, I will highlight two such features.

*6.1. Ephesians 4:9–12 and the Narrative of Acts*

The material in Eph 4:1—6:20 is frequently categorized as "paraenesis" in contrast with the "doctrinal" material in 1:3—3:21 (e.g., Best 1998, p. 353). While this description

is broadly appropriate, it does not explain every feature of 4:1—6:20. Most significantly, 4:4–16 seems to break away from the paraenetic form to introduce new material concerning the "body" of the *christos*, i.e., the church/*ekklēsia* (cf. 1:22–23) (Best 1998, p. 354). What is the purpose of this non-paraenetic description of the body of the *christos*?

As several interpreters argue, the "descent" (4:9–10) may refer to Jesus' coming to his people at Pentecost in the person of the Holy Spirit (e.g., Caird 1976, pp. 73–75; Fowl 2012, pp. 138–40; Lincoln 1990, pp. 244–47).[34] This view is sometimes rejected because the Holy Spirit is not mentioned explicitly (Best 1998, p. 386). However, the focus of the discourse at this point is on the *christos* as the ascended and descended giver of gifts (4:7–10). Furthermore, the ascended *christos* has previously been described as the primary actor in the apostolic mission (2:17, cf. 2:6). Hence, it is entirely plausible that the ascended *christos* would also be described here as the primary actor in the redemptive-historical moment at which the apostolic mission began—i.e., Pentecost.

This suggests that Eph 4:9–12 may be read coherently as a descriptive narrative recalling concrete events surrounding the progress of the apostolic mission from Israel to the nations. At Pentecost, the victorious ascended *christos* Jesus descended to his people in the person of the Spirit (4:9–10; cf. Acts 2, esp. 2:38). The *christos*, through the Spirit, gave key figures (apostles, prophets, etc.) to the original Israelite apostolic community (4:11). Consequently, these Israelite "holy ones" were restored and equipped with a ministry towards the more expansive "body" of the *christos* (4:12).[35] Following this, 4:13–16 depicts the projected future goal of the *ekklēsia* as a mature "body" (cf. 1:22–23; 3:10, 21). As Korner (2017) has argued, since the term *ekklēsia* was used by Jews as both a supra-local national identity (the *ekklēsia* of Israel in the desert wanderings) and a local group designation in Alexandria (Philo, *Virt.* 108), Pauline use of this term along with associated metaphors such as "body" would have served to bind Pauline *christos*-followers more closely with the Jerusalem apostolic community.

Thus, 4:7–16 is not simply paraenetic material providing an ideal blueprint for church structures in every generation. Instead, it is descriptive material narrating *how* the ascended Jewish *christos*, through the apostolic mission from Israel to the nations, has built and is building his body, the church/*ekklēsia* (cf. 1:20–23). Nevertheless, the descriptive material serves a paraenetic purpose, emphasized at the beginning and end of the section: it affirms that in the body of the *christos*, diversity—including Jew-gentile diversity—remains a positive element supporting and enabling unity through the mutual sharing of blessings (4:7, 16).

### 6.2. Gentile Halakhah in the Jewish Christos/Messiah

Another question concerns the basis for the paraenesis in Eph 4—6. While much of the material is christologically grounded (e.g., 4:20, 32; 5:2, 14, 21–32; 6:5), the Mosaic law also features, both implicitly (e.g., 4:24; cf. Gen 1:26–28) and explicitly (e.g., 5:31, 6:2–3; cf. Gen 2:24; Exod 20:12). What is the relationship of the paraenetic material to the Mosaic law? Two considerations help to answer this question. Firstly, in Eph 1—3, the readers are depicted as "gentiles" who have been included in the holiness of Israel. Secondly, the term "walk" (περιπατέω) is prominent as a description of the paraenesis (4:1, 17; 5:2, 8, 15; cf. 2:2, 10). This is a distinctly Pauline usage deriving from the Hebrew scriptural term *halakh* (הָלַךְ) (Seesemann and Bertram 1967). This suggests that the paraenesis of Eph 4:1—6:20 is being framed as a christologically grounded form of gentile *halakhah*. Its purpose is to apply key elements of Israel's law to the readers' situation as gentiles redeemed in the Jewish *christos*. These gentiles are "no longer to walk as the gentiles walk" (4:17) but rather to "walk" in light of their calling to be united with Israel in the *christos* (4:1–6).

### 7. An Evangelical Post-Supersessionist Reading of Colossians

While Jewish elements are present in Colossians, they are less prominent than in Ephesians. Hence, the discussion of Colossians presented here will be briefer.

### 7.1. Colossians and the Apostolic Mission

Several features of Colossians are similar to those found in Ephesians and can be understood on the basis that the letter is situated within the apostolic mission proceeding from Israel to the nations. These features include the situating of the Colossians' faith within the worldwide apostolic gospel mission (Col 1:5–7; cf. Eph 1:13–14), the depiction of the recipients as being qualified "for the portion of the inheritance of the holy ones" (εἰς τὴν μερίδα τοῦ κλήρου τῶν ἁγίων) (Col 1:12; cf. Eph 1:11, 14, 18), and the depiction of Paul's ministry in terms of a "mystery" (μυστήριον) which was "revealed to his [God's] holy ones" (ἐφανερώθη τοῖς ἁγίοις αὐτοῦ) (Col 1:26; cf. Eph 3:3–5) and whose riches are shared "among the nations" (ἐν τοῖς ἔθνεσιν) (Col 1:27; cf. Eph 3:6). There is also a marked reference to the Jewish identity of Paul's fellow-workers in the apostolic mission, who are singled out because they are "from the circumcision" (ἐκ περιτομῆς) (Col 4:11).

### 7.2. The Nature of the Threat

However, there is a difference in focus between Ephesians and Colossians. Whereas Ephesians often highlights the broad scope of the apostolic mission from Israel to the nations, Colossians is more focused on an immediate threat to its readers. The nature of the threat, and its relation to possible Jewish elements, has been the subject of much debate. Many interpreters regard the threat as arising from concrete Jewish rivals (e.g., Bevere 2003, pp. 53–147; Bird 2009, pp. 15–26; Dunn 1996, pp. 29–33; Wright 1986, pp. 23–33). Although this may account for some elements, such as circumcision (2:11; 3:11) and sabbaths (2:16), many of the issues addressed in Colossians are not easily confined to a specifically Jewish setting. Hence, other interpreters see an entirely non-Jewish threat arising from an ascetic Hellenistic philosophy (e.g., Allen 2018; Martin 1996). While this accounts for much of the data, it does not explain the mention of circumcision in 2:11. It is best, therefore, to follow those interpreters who regard the threat as arising from a combination of Hellenistic and apocalyptic Jewish thought with a focus on ascetic mysticism (e.g., Arnold 1996; Beale 2019, pp. 12–16; Foster 2016, pp. 10–16). In this view, negative references to Jewish elements do not necessarily constitute opposition to Judaism per se; they are simply part of the broader opposition to the ascetic mystical ideas threatening the Colossians' faith. Therefore, the presence of a negative evaluation of a Jewish element in Colossians does not necessarily entail a supersessionist viewpoint. Instead, each reference should be assessed on a case-by-case basis.

### 7.3. Circumcision (Colossians 2:11–13)

Colossians 2:11–13 affirms a "non-hand-made circumcision" (περιτομῇ ἀχειροποιήτῳ) (2:11) given to gentiles who believe in Jesus as the *christos* (cf. 2:6). As we have seen, this has led some interpreters to infer that the author regards physical circumcision as entirely devoid of value, even for Jewish believers. This conclusion arises from the view that there is an implicit contrast between physical circumcision and non-physical circumcision. However, this is simply an assumption since physical circumcision is not explicitly mentioned in this passage. It is entirely plausible that the implicit contrast is with an *alternative non-physical circumcision* offered by the ascetic religious philosophy. The idea of a non-physical circumcision for gentiles is found elsewhere (Philo, *QE* 2.2), so it may have been a feature of the religious philosophy in Colossae. If so, the problem addressed in 2:11–13 is not physicality; instead, the issue is that the alternative spiritual experience is not grounded in the *christos*. While this cannot be demonstrated with certainty, it does show that it is unnecessary to assume that Col 2:11–13 opposes physical circumcision for Jews.

### 7.4. The Mosaic Law (Colossians 2:13–23)

Colossians 2:13–23 describes the death of the *christos* in opposition to various entities that some interpreters have linked with the Mosaic law. As we have seen, this has led these interpreters to infer that the author regards the *christos* as having abolished any concrete Jewish observance of the Mosaic law.

The first entity is "the handwritten record in/with the decrees" (τὸ ...χειρόγραφον τοῖς δόγμασιν) that is canceled (2:14). This is almost certainly an eschatological record of debt arising from the recipients' "trespasses" (παραπτώματα) (2:13). While these "trespasses" may (or may not) have been against the Mosaic law, there is no need to regard the *christos* here as abolishing the law itself. Bevere (2009) argues that this passage should be understood in light of the parallel in Eph 2:15, which refers to the abolition of "the law of the commandments in decrees." However, even if the parallel is valid, as we have seen, Ephesians does not imply the abolition of concrete Jewish observance of the Mosaic law (see above).

The second entity is "the elements of the world" (τὰ στοιχεῖα τοῦ κόσμου) that threaten to regulate the recipients' lives (2:8, 20). The term "elements" has a range of meanings in ancient texts (Martin 2018). In this context, the phrase most likely refers to religious observances associated with the natural world (Foster 2016, pp. 252–54). Dunn (1996, pp. 150–51) argues that this may be linked to the festivals of the Mosaic law, citing the parallel expression in Gal 4:3, 9. However, even in Galatians, the association between the "elements of the world" and the calendrical observances of the Mosaic law is at most indirect (De Boer 2007). Hence, the parallel is insufficient grounds to see a clear reference to the Mosaic law in Colossians. Indeed, the phrase may be grounded in Jewish critiques of Stoic or other Hellenistic notions of deity (cf. Wis 13:1–4) (Engberg-Pedersen 2010, pp. 90–92).

The third entity is food and calendrical observances including "festival[s]," "new moon[s]," and "Sabbaths" (2:16) that are described as a "shadow of the coming things" (σκιὰ τῶν μελλόντων), seemingly opposed to the "body [which is] of the *christos*" (τὸ δὲ σῶμα τοῦ χριστοῦ) (2:17). Several interpreters (e.g., Dunn 1996, pp. 176–77) see the "shadow"-"body" metaphor as a reference to a salvation-historical supersession. In this view, the "shadow" denotes Jewish law-observances in the time before the *christos* (cf. Heb 10:1), while the "body" is the fulfillment of the law that has now arrived in the *christos*. Since the shadow was only an outline designed to prefigure the *christos*, it is now superseded. However, in Colossians, the "shadow"-"body" metaphor is used in a far more oppositional sense (cf. Josephus, *J.W.* 2.28; Philo, *Heir* 72). The "shadow" is not assigned any positive value, even as an outline or pointer to the *christos*; instead, the metaphor seems to denote an insubstantial and ineffective mimicry. Furthermore, "the coming things" (τῶν μελλόντων) most likely has a more thoroughgoing eschatological referent; i.e., it refers not to the present but the future appearing of the *christos* (3:4; cf. Rom 8:18, 38; 1 Cor 3:22; Eph 1:21; 1 Tim 4:8; 6:19; 2 Tim 4:1). Hence, it is best to understand this passage simply to be claiming that the ascetic religious practices that promised spiritual experiences are at best an ineffective mimicry of believers' future glory in the *christos* and so should be abandoned (Foster 2016, pp. 283–85). Again, while the ascetic practices may have incorporated Jewish elements, it is unnecessary to assume that the author is opposing Jewish practices per se.[36]

### 7.5. *Jewish Identity (Colossians 3:9–11)*

Colossians 3:9–11 describes a "new" kind of "humanity" in which "there is not Greek and Jew, circumcision and uncircumcision, barbarian, Scythian, slave, free, but all things and in all things [is] *christos*" (οὐκ ἔνι Ἕλλην καὶ Ἰουδαῖος, περιτομὴ καὶ ἀκροβυστία, βάρβαρος, Σκύθης, δοῦλος, ἐλεύθερος, ἀλλὰ τὰ πάντα καὶ ἐν πᾶσιν χριστός) (3:11). As we have seen, this has led some interpreters to infer that the author regards Christ as having negated all forms of distinct Jewish identity. However, this passage may be understood similarly to our reading of the "one new humanity" in Eph 2:15 (see above). The point is that there is a new sphere of existence brought about by the *christos* in which human social distinctions are no longer the basis for enmity or divisiveness (cf. Col 3:8). This does not imply that distinct social identities are entirely invalid. Indeed, given the positive reference to the circumcision of Paul's missionary co-workers in 4:11, it would be an illegitimate over-reading to regard 3:11 this way.

## 8. Conclusions and Implications

In this article, I have presented a post-supersessionist reading of Ephesians and Colossians, summarizing the arguments from my book *Reading Ephesians and Colossians After Supersessionism* (Windsor 2017). My reading strategy involves seeking to read the letters as situated within the dynamics of the apostolic mission to proclaim the gospel of Jesus as the Jewish *christos*/messiah to the nations. In Acts, this apostolic mission is described as a dynamic in which the blessings of salvation in the *christos* begin within a distinctly Israelite original community and proceed to the nations. While these blessings fundamentally transform the nature of Jewish and gentile identity, they do not entirely negate the value of Israelite distinctiveness. I have argued that there is a strong warrant for situating Ephesians and Colossians within a similar dynamic. I have highlighted significant instances where this "priestly" dynamic appears in the letters. I have also sought to demonstrate how this dynamic provides satisfactory answers to specific exegetical questions. Furthermore, I have offered alternative non-supersessionist readings of critical passages concerning circumcision, the Mosaic law, and Jewish identity that do not negate the value of Jewish distinctiveness in the *christos*.

The focus of this study is exegetical. Hence, the study does not address contemporary religious, theological, missiological, ecclesiological, and hermeneutical questions in detail. Nevertheless, the reading is offered in the hope that highlighting aspects of these letters that are often neglected will stimulate further reflection in these areas. Rather than simply viewing the letters as compendiums of abstract theological and ethical pronouncements, this reading highlights the dynamic salvation-historical contours that provide a rich setting and rationale for their theological and ethical expressions. Rather than viewing the concept of "unity" as a totalizing concept that seeks to eradicate all distinctions, this reading highlights various ways in which an appropriate affirmation of differences and distinctions, especially ethnic distinctions, can perform a profoundly positive function in relation to unity in Jesus as the *christos* (cf. 1 Cor 7:17–24) (Tucker 2011). Rather than seeing ethnic difference only as a cause for hostility, this reading highlights how ethnic difference—in particular, here, the distinctiveness of Jew and gentile—can be an instrument for mutual service through the communication of divine blessing, grounded in the gospel of Jesus as the Jewish *christos* for the nations.

**Funding:** This research received no external funding.

**Conflicts of Interest:** The author declares no conflict of interest.

## Notes

[1] The term "post-supersessionism" is defined in Soulen (2005).

[2] The ensuing content in its revised form is used with permission (www.wipfandstock.com, accessed on 19 October 2022). I have also included some more recent scholarship and made some minor updates.

[3] Compared with Ephesians, there is less explicit post-supersessionist interpretation of Colossians. Allen (2018) is an exception.

[4] See also other works by Barth (1960; 1969, pp. 79–117; 1983, pp. 45–49).

[5] In this context, "evangelical" does not denote a confessional commitment, although the interpretive approach outlined is consistent with such a commitment.

[6] This does not necessarily assume historical Pauline authorship of the letters. I am approaching Colossians and Ephesians as documents aligned with concerns evident in Acts—which was not written by Paul—and in Romans—which was written by Paul. Nevertheless, the findings presented here are consistent with historical Pauline authorship and might provide evidence in its favor.

[7] Cf. Campbell (2014, pp. 254–338), who provocatively argues for an early (50 CE) date for Philemon, Colossians, and Ephesians and locates them within the historical contingencies of Paul's mission. According to Campbell, Paul wrote all three letters during an imprisonment in Asia near the east of the Lycus valley. Ephesians was originally the letter to the Laodiceans (so Marcion's text of Eph 1:1; cf. Col 4:16). Paul wrote it to new converts whom he knew about but had not met, for the purpose of "construction" of their "Christian identity" (p. 325). Viewed this way, Ephesians (or "Laodiceans") provides "a relatively straightforward account of Paul's missionary agenda in relation to pagan conversion" (p. 314). The material on Jew-gentile relations in the letter fits the issues raised just previously in meetings at Syrian Antioch and Jerusalem (pp. 329–30). Even if we do not accept all of Campbell's

(controversial) conclusions, his argument exemplifies how many details of both Colossians and Ephesians can be read in a way that is plausibly consistent with a location within Paul's missionary endeavors rather than outside them.

8    The aorist indicative (i.e., past tense) is the default verbal form for narratives. The existence of such narrative constructions in Ephesians and Colossians suggests that the author is not simply describing timeless theological truths but locating the readers within a shared history.

9    This argument does not depend on an early date for Acts, Ephesians, or Colossians. Acts, whether written early or late, *portrays* the progress of the apostolic mission as an Israel-centric dynamic. Ephesians and Colossians, whether authentically Pauline or deutero-Pauline, display a range of features that align with this perspective found in Acts. This is evidence that the non-supersessionist perspective in Acts is shared by the author(s) of Ephesians and Colossians.

10   The terminology includes "the word" (ὁ λόγος) (Eph 1:13; Col 1:5; Acts 10:36, 44; 11:1) "gospel"/"evangelize" (εὐαγγέλιον/ εὐαγγελίζω) (Eph 1:13; Col 1:5, 23; Acts 10:36), "hear" (ἀκούω) (Eph 1:13; Col 1:6, 23; Acts 10:22, 33, 44), "faith"/"believe" (πίστις/πιστεύω) (Eph 1:13; Col 1:4, 23; Acts 10:43; 11:17), "salvation"/"save" (σωτηρία/σῴζω) (Eph 1:13; Acts 11:14), "the Holy Spirit" (τὸ πνεῦμα ...τὸ ἅγιον) (Eph 1:13; Acts 10:44–45, 47; 11:15), and "glorify"/"glory" (δοξάζω/δόξα) (Eph 1:14; Acts 11:18).

11   While Lopez (2008) presents an impressive array of evidence that the phrase "the nations" would have been understood by the average inhabitant of the Roman Empire as a reference to people groups violently conquered by the Empire, Ephesians pointedly defines the phrase with reference to Israel and explicitly calls on readers to consider themselves from that viewpoint (Eph 2:11–13). Colossians also describes "the nations" in relation to "the holy ones" (Col 1:26). Thus, the usage in Ephesians and Colossians is more consistent with the Israel-centric use of the term in Acts than with the political understanding highlighted by Lopez.

12   Several of these commonalities between Ephesians and Acts were noted earlier by Käsemann (1968). Käsemann saw these connections as demonstrating that Ephesians "most clearly marks the transition from the Pauline tradition to the perspectives of the early Catholic era" (288). Interestingly, over the last half-century, prevailing views concerning the differences between Ephesians and the undisputed Pauline epistles on supersessionism have reversed. Käsemann (1968) argued that the historical Paul was supersessionist, but the author of Ephesians, along with Acts, was non-supersessionist (pp. 296–97). By contrast, Lincoln (1990) argues that the historical Paul was non-supersessionist, but the author of Ephesians was supersessionist (xcii–xciii). The fact that these views are opposed highlights the extent to which prevailing presuppositions can influence scholarly pronouncements concerning supersessionism in disputed and undisputed Pauline epistles, underlining the need for careful and nuanced reading.

13   Korner (2020, pp. 185–87) observes that the phrases used in Eph 2:20 (τῶν ἀποστόλων καὶ προφητῶν) and 3:5 (τοῖς ἁγίοις ἀποστόλοις αὐτοῦ καὶ προφήταις) do not include the article before the second noun and so may be read as a hendiadys: "apostle-prophets." Conversely, the expression used in Eph 4:11 (τοὺς μὲν ἀποστόλους, τοὺς δὲ προφήτας) includes both an article (τούς) and a development marker (δέ) before the second noun and so must be referring to two separate groups "the apostles, the prophets, etc." Korner suggests that there are two groups: "Ephesians presents the first group ('apostle-prophets') as being foundational to the universal *ekklēsia* (2:20; 3:5) while the second set [i.e., apostles and prophets] (4:11) fulfill similar functions but for regional *ekklēsiai* without the attendant spiritual authority characteristic of the first" (p. 187). I follow a different line of interpretation regarding Eph 4:11, seeing it as a further reference to key figures in the original Israelite apostolic community (see below). Hence, following Sandnes (1991, pp. 234–36), I regard the "apostles" and "prophets" in all three places (Eph 2:20; 3:5; 4:11) as non-identical yet closely related—and possibly overlapping—groups who are foundational to the apostolic mission. Nevertheless, on either understanding, Eph 2:20 highlights the prophetic authority of the apostles and the ongoing foundational relevance of the original Israelite apostolic community for the gentile *ekklēsiai*.

14   I have made a more detailed comparison of Ephesians and Barnabas elsewhere (Windsor 2018).

15   Buell (2005) surveys racial/ethnic reasoning in early Christian theology, demonstrating that it was far more complex and multi-faceted than the modern idea of "replacement" might suggest.

16   The KJV translates the original of Eph 2:15 (ἵνα τοὺς δύο κτίσῃ ἐν αὐτῷ εἰς ἕνα καινὸν ἄνθρωπον) fairly literally as "for to make in himself *of twain* one new man." The RSV updates this to read: "that he might create in himself one new man *in place of the two*."

17   $\mathfrak{P}^{46}$ (which omits τοῖς), ℵ*, B*, 6, 424$^c$, 1739, etc.

18   ℵ$^2$, A, B$^2$, D, F, G, 33, 81, etc.

19   Rom 1:7; 1 Cor 1:2; 2 Cor 1:1; Phil 1:1.

20   The suggestions listed by Best (1997a, pp. 4–5) do not explain the resulting grammatical awkwardness. The suggestion that this was originally a circular letter with a space to write different destinations is "conjectural" and has "considerable difficulties" along with other conjectures (Best 1997a, p. 10).

21   Cf. Campbell's (2014, pp. 309–38) argument that the letter was originally intended for Laodicea (see above).

22   See Gen 9:26; 24:27; 1 Sam 25:32; 1 Kgs 1:48; 8:15; 1 Chr 16:36; 29:10; 2 Chr 2:12; 6:4; Ezra 7:27; Pss 41:13; 68:35; 72:18; 106:48; Tob 8:5; 1 Macc 4:30; 3 Macc 7:23; 1QM 13.2; Luke 1:68.

23   There is also a transition from "you" (ὑμῖν) to "us" (ἡμᾶς) in 1:2–3.

24   See also (Caird 1976, p. 41; Cohick 2010, p. 52; Starling 2011, pp. 186–89). Kinzer (2015, pp. 69–73) and Campbell (2008, p. 22) regard the reference to Israel as extending as far back as the beginning of the blessing (1:3).

25    On the close connection (if not identity) between apostles and prophets, see note 13.

26    The grammatical apposition of "the dividing wall" (τὸ μεσότοιχον) and "the hostility" (τὴν ἔχθραν) implies that the two phrases have the same referent. The simplest explanation for this is that the dividing wall is a metaphor for the hostility.

27    The reference to the "yoke that neither our fathers nor we were able to bear" (Acts 15:10) is difficult, but it need not be understood as circumcision per se. The issue in this context is not circumcision itself, but *gentile* circumcision as a requirement for *eschatological salvation*. Hence the "yoke" may simply be a reference to an impossibly strict interpretation of the Law requiring gentile circumcision.

28    In 2:5–6, the readers are described as "made alive together" (συνεζωοποίησεν), "raised together" (συνήγειρεν), and "seated together" (συνεκάθισεν) with the *christos*. The use of the *syn*-compounds indicates that believers share in the status of the risen *christos*, not that they have merged with or replaced the *christos*.

29    Elsewhere, depending on the context, the phrase "the holy ones"/"the saints" (οἱ ἅγιοι) can refer either to the original apostolic community in Jerusalem (Acts 9:13; Rom 15:25–26, 31; 1 Cor 16:1; 2 Cor 8:4; 9:1, 12) or to all believers (e.g., Phil 1:1) (cf. Trebilco 2012, pp. 122–63). Since the point here is that gentile believers *share* in the holiness of the original apostolic community, it is difficult to decide which meaning fits best here.

30    I am grateful to Kennedy (2018), who in reviewing my published book noted that I had missed this allusion.

31    Cf. Staples (2011) who argues similarly concerning Rom 11:25–27.

32    The genitive construction "the foundation *of* the apostles and prophets" (τῷ θεμελίῳ τῶν ἀποστόλων καὶ προφητῶν) is best understood as a genitive of source, i.e., the foundation *laid by* the apostles and prophets through their gospel preaching (Sandnes 1991, pp. 227–29).

33    This phrase has previously been defined in explicitly Israel-centric terms (2:11–13); cf. note 11.

34    An early interpretation regards this as a reference to Jesus' post-crucifixion descent to Hades, either to conquer Satan, to proclaim the gospel to the dead, or to draw faithful departed Israelites to himself before ascending (e.g., Tertullian, *An.* 55.2; cf. the addition of "first" (πρῶτον) after "he descended" (κατέβη) in several witnesses such as ℵ², B, C³, K, L) (see Thielman 2010, pp. 268–72). Calvin (1965, p. 176) saw it as Jesus' incarnation, leading to his humility death; this is followed by many modern interpreters (e.g., Barth 1974, vol. 2, pp. 433–34; Best 1998, pp. 383–86; Hoehner 2002, pp. 533–36). While these possibilities have an impressive pedigree, the reading presented in this article also has significant support, with the added benefit that it is more closely integrated with the concerns of the discourse concerning gift-giving and the *ekklēsia* as the "body" of Christ in Eph 4.

35    As noted above, depending on the context, the phrase "the holy ones"/"the saints" (οἱ ἅγιοι) can refer either to the original apostolic community in Jerusalem or to all believers who share in their holiness.

36    Allen (2018) has offered an alternative post-supersessionist interpretation worthy of careful consideration. In Allen's view, the author has a positive view of specifically *Jewish* food practices and calendrical observances (cf. Martin 1996, pp. 124–34). The author urges the recipients not to allow adherents of non-Jewish ascetic religion to judge them for following such observances. This is because these observances are a (positive) shadow/outline of the future eschatological kingdom. Rather than being judged by others for their participation in the Jewish festivals, the recipients should pay due regard to "the corporate body of messiah" (p. 143). While this interpretation is possible, it does not fully explain the very close grammatical parallel between "shadow" and "body" (2:17). This parallel appears to be invoking a common double-sided metaphor. Furthermore, it does not easily account for the negative mention of "circumcision" in 2:11 which suggests that there were at least some Jewish elements in the religious philosophy.

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
