# Peer review of "Israel and the Apostolic Mission: A Post-Supersessionist Reading of Ephesians and Colossians"

_religions, doi:10.3390/rel14010044_

Round 1

Reviewer 1 Report

This is an excellent piece of academic writing. Rooted in a close and careful reading of the select biblical passages, it makes a convincing case for an evangelical post-supersessionist reading. It seriously engages with theological literature, both classic and recent (Christian and, to a certain extent, Jewish). The present reviewer recommends publishing this manuscript.

Author Response

I am grateful for your review, thank you. I appreciate your recommendation to publish.

Reviewer 2 Report

The aim of the paper is clear, the title relevant. The author, starting from the supersessionist interpretation of Ephesians and Colossians, intends to propose a new reading of them. The excusatio on lines 36-39 is superfluous. It’s very clear what is already known about the topic. The author outlines clearly the previous research, to purpose his new reading.

Author Response

I am grateful for your review, thank you. I appreciate your comment about the excusatio on lines 36–39. I have retained this sentence in the resubmitted version here but I am very happy for the Special Edition editor to remove this sentence if he prefers to do so.

Reviewer 3 Report

This is a high-quality essay that offers strong post-supersessionist readings of passages within Ephesians and Colossians. The arguments are interesting and well-argued, interacting with secondary literature when helpful. My main concern is that the author acknowledges that it is a summary of the arguments from a previously published book. I leave it up to the editors of this issue to decide if that genre of work fits with the issue. Perhaps a book review of the book instead? I normally think of journal articles as pushing forward new ideas that haven't been tested yet, not summarizing already published work. But as I said, the essay is of good quality. 

Author Response

I am grateful for your review, thank you. I appreciate your concern about the article being a summary of the book. I had pre-checked this with the Special Editions editor and I understand that he is fine with this. I will leave it to him to take your comments into account.